# MACHINE LEARNING FOR PROTAC ENGINEERING ICRL 2024

## ABSTRACT

PROTACs are a promising therapeutic technology that harnesses the cell's built-in degradation processes to degrade specific proteins. Despite their potential, developing new PROTAC molecules is challenging and requires significant expertise, time, and cost. Meanwhile, machine learning has transformed various scientific fields, including drug development. In this work, we present a strategy for curating open-source PROTAC data and propose an open-source toolkit for predicting the degradation effectiveness, *i.e.*, activity, of novel PROTAC molecules. We organized the curated data into 16 different datasets ready to be processed by machine learning models. The datasets incorporate important features such as $pDC_{50}$, $D_{max}$, E3 ligase type, POI amino acid sequence, and experimental cell type. Our toolkit includes a configurable PyTorch dataset class tailored to process PROTAC features, a customizable machine learning model for processing various PROTAC features, and a hyperparameter optimization mechanism powered by Optuna. To evaluate the system, three surrogate models were developed utilizing different PROTAC representations. Using our automatically curated public datasets, the best models achieved a 71.4% validation accuracy and a 0.73 ROC-AUC validation score. This is not only comparable to state-of-the-art models for protein degradation prediction, but also open-source, easily reproducible, and less computationally complex than existing approaches.

## 1 INTRODUCTION

Machine learning (ML) has transformed various scientific domains, including drug design and discovery, by offering novel solutions to complex, multi-objective optimization challenges (Nori et al., 2022). In the context of medicinal chemistry, ML techniques have revolutionized the process of identifying and optimizing potential drug candidates. Traditionally, drug discovery has relied heavily on trial-and-error experimentation, which is not only time-consuming but also expensive. ML techniques have the potential to significantly accelerate this process by predicting properties of molecules *in silico*, such as binding affinity, solubility, and toxicity, with remarkable accuracy (Mercado et al., 2021; Blaschke et al., 2020). This in turn saves time and money to focus on the most promising candidates.

In order to develop ML models for chemistry, ML algorithms leverage vast datasets containing molecular structures, biological activities, and chemical properties to learn intricate patterns and relationships, also called quantitative structure-activity relationships (QSAR). These algorithms can discern subtle correlations and structure in molecular data that are difficult for human experts to identify. Consequently, ML-based approaches aid in predicting which molecules are likely to be effective drug candidates, thereby narrowing down the search space and saving resources.

PROTACs, or PROteolysis TArgeting Chimeras, represent an innovative class of therapeutic agents with immense potential in challenging disease areas (Liu et al., 2020; Tomoshige & Ishikawa, 2021; Hu & Crews, 2022). Unlike traditional small molecule inhibitors, PROTACs operate by harnessing the cell's natural protein degradation machinery, the proteaosome, to eliminate a target protein of interest (POI). This catalytic mechanism of action towards targeted protein degradation offers several advantages over conventional approaches, which frequently work by having a small molecule drug bind tightly to and thus block a protein's active site. By selectively degrading the target protein, PROTACs offer a more comprehensive effect at potentially lower doses compared to traditional

inhibitors. This approach is particularly relevant in cases where inhibiting target activity might not be sufficient; notable examples include certain neurodegenerative diseases like Alzheimer's, where misfolded proteins agglomerate and lead to negative downstream effects in patients (Békés et al., 2022).

Because of their unique mechanism, which does not require tight binding to well-defined protein pockets, PROTACs provide a promising avenue for tackling previously "undruggable" targets, thus expanding the realm of therapeutic possibilities and opening the doors to new treatment strategies. This innovative approach to protein modulation underscores the importance of understanding the protein degradation capabilities of PROTACs in order to advance their development to large-scale clinical trials. Nevertheless, despite the potential of PROTACs, there is a notable scarcity of open-source tools and resources tailored specifically for working with this class of molecules (Mostofian et al., 2023). Existing software tools and datasets for general drug discovery engineering focus almost exclusively on small molecule inhibitors, leaving a gap in methods applicable to the PROTAC modality in terms of research and analysis. This gap poses a considerable hurdle for researchers and developers seeking to explore the full potential of PROTACs, one which we seek to address via the open-source tools for working with PROTACs presented in this work.

## 2    CONTRIBUTIONS

We summarize our contributions to the field of PROTAC engineering as follows:

- The meticulously curation and collection of open-access PROTAC data, sourced from PROTAC-Pedia (London, 2022) and PROTAC-DB (Weng et al., 2021), facilitating comprehensive data-driven analysis.
- The provision of an open-source toolkit that includes essential scripts and tools for effective manipulation and exploration of PROTAC data, catering to both public domain and privately held repositories.
- The development of advanced deep learning models for accurate protein degradation prediction from intuitive molecular representations, achieving a validation accuracy comparable to existing more complex baselines (Li et al., 2022).

## 3    BACKGROUND

PROTACs, short for PROteolysis TArgeting Chimeras, have emerged as a novel approach to drugging challenging protein targets (Li & Crews, 2022). Unlike the majority of traditional small molecule inhibitors, PROTACs operate through a distinct mechanism of action: targeted protein degradation via induced proximity. These chimeric molecules are designed to recruit a specific target protein and an E3 ubiquitin ligase, leading to the target protein's ubiquitination and subsequent proteasomal degradation within the cell. This unique mode of action enables PROTACs to effectively remove disease-relevant proteins from the cellular environment, offering a novel strategy for therapeutic intervention.

Assessing the efficacy of PROTAC molecules is a critical aspect of their development, and measurement of dose-response curves is commonly employed for this purpose (Gesztelyi et al., 2012). The dose-response curve involves first exposing cells to varying concentrations of a PROTAC, then measuring the resulting reduction in the target protein's expression (presumably, because it has been degraded). This approach helps establish the concentration required for half-maximal degradation ($DC_{50}$) and maximum degradation ($D_{max}$), two crucial metrics for evaluating a PROTAC's potency.

Critical to understanding dose-response curves is the *hook effect* (Semenova et al., 2021), a phenomenon where extremely high concentrations of PROTACs can lead to reduced degradation efficacy. This effect arises due to the saturation of the cellular degradation machinery, resulting in diminished degradation despite increased PROTAC concentrations. Recognizing and addressing the hook effect is essential to accurately interpreting degradation data and avoiding potential misinterpretations during the evaluation process.

Despite the promising potential of PROTACs, the availability of high-quality, open datasets for training and validation of degradation models remains a challenge (Mostofian et al., 2023). Existing

datasets often suffer from limitations in terms of size, diversity, and data quality. On top of that, many datasets are specific to small molecule inhibitors, making them ill-suited for training models tailored to PROTACs. Moreover, the scarcity of comprehensive and curated datasets for PROTAC molecules hinders the development of accurate predictive models. This scarcity of suitable data impedes progress in PROTAC research and limits researcher's ability to fully harness the potential for targeted protein degradation via PROTAC engineering. In this work, we address many of the aforementioned limitations and introduce a Python toolkit[1] designed to facilitate PROTAC research through curated datasets and advanced deep learning models.

## 4 RELATED WORK

To the best of our knowledge, PROTAC-DB (Weng et al., 2021) and PROTAC-Pedia (London, 2022) are the largest openly-available datasets that include PROTAC data. PROTAC-DB contains structural and experimental data of around four thousand PROTAC complexes along with web-scraped data from the scientific literature. While the PROTAC-DB database allows users to query, filter, and analyze PROTAC data, and to compare different compounds based on parameters such as $DC_{50}$ and $D_{max}$ via its online platform, its data is not specifically structured for ML models, but rather for online access through its web page. Wrangling the data for use in data-driven models requires significant cleaning and curation.

On the other hand, PROTAC-Pedia provides crowd-sourced entries with details about PROTAC molecules and their degradation activity. However, its definition of degradation activity is less precise compared to PROTAC-DB. In fact, while in PROTAC-DB a PROTAC molecule is deemed active when its $D_{max}$ exceeds 80% and its $DC_{50}$ remains below 0.1 $\mu$M (equivalently, a $pDC_{50} > 7$), PROTAC-Pedia entries are marked active when their $D_{max}$ surpasses 30% and their $DC_{50}$ stays below 100 $\mu$M (equivalently, a $pDC_{50} > 5$).[2]

The studies most closely aligned with our work are those of Li et al. (2022) and Nori et al. (2022). Li et al. (2022) introduces DeepPROTACs, a deep learning model for prognosticating PROTAC activity, whereas Nori et al. (2022) introduces a LightGBM model for predicting protein degradation activity. LightGBM is a gradient boosting framework that uses a histogram-based approach for efficient, high-performance ML tasks Ke et al. (2017).

The DeepPROTACs architecture encompasses multiple branches employing long short-term memory (LSTM) and graph neural network (GNN) components, all combined prior to a prediction head. Each branch processes distinct facets of the PROTAC-POI complex, encompassing elements like E3 ligase and POI binding pockets, along with the individual components of the PROTAC: the warhead, linker, and E3 ligand. The model's performance culminates in an average prediction accuracy of 77.95% and a ROC-AUC score of 0.8470 on a validation set drawn from the PROTAC-DB. The LightGBM model, on the other hand, achieves a ROC-AUC of 0.877 on a PROTAC-DB test set with a much simpler model architecture and input representation.

Notwithstanding their achievements, the DeepPROTACs and LightGBM models both exhibit certain limitations. In DeepPROTACs, there is a potential risk of information loss as the PROTAC SMILES are partitioned into their constituent E3 ligands, warheads, and linkers, which are then fed into separate branches of the model. Secondly, while the authors undertake advanced molecular docking of the entire PROTAC-POI-E3 ligase complex, their subsequent focus on the 3D binding pockets of the POI and E3 ligase renders it less amenable for experimental replication and practical use. Thirdly, the model's performance is not benchmarked against other SOTA models with high predictive capacity, such as LightGBM or XGBoost. Finally, and perhaps most importantly, the potential for data leakage during hyperparameter optimization and its effects on out-of-distribution (OOD) generalization was not investigated. Data leakage between the different PROTAC components in the training and test sets of the model may artificially render a more accurate model that does not generalize well to new real-word data, necessitating more rigorous testing procedures. Because of that, generalization of the DeepPROTACs model would need to be further investigated on a separate validation set.

---

[1] https://anonymous.4open.science/r/protac-toolkit
[2] $DC_{50}$ is measured in molar units "M", whereas $pDC_{50}$ is expressed as negative "$Log_{10}(M)$" units.

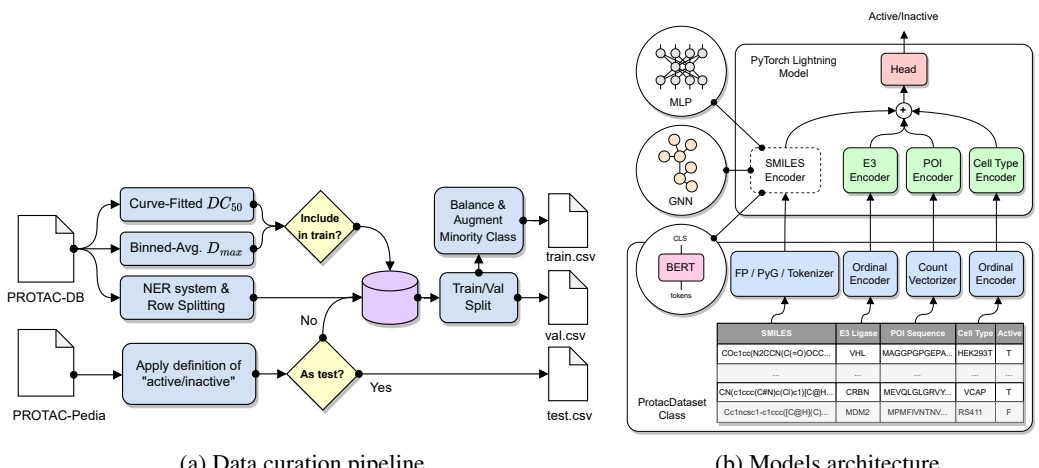

(a) Data curation pipeline                         (b) Models architecture.

Figure 1: An overview of (a) the data curation pipeline and (b) the models constructed in this work.

# 5 METHODS

This section provides an overview of our methodology in curating the available open-source PROTAC datasets, the assembly of 16 different ML-ready datasets from this highly-curated data, and a description of various proposed machine learning models for targeted protein degradation prediction. An overview of our data curation pipeline is illustrated in Figure 1a, whereas the proposed model architectures are shown in Figure 1b.

## 5.1 DATA CURATION

To prepare the data for our models, we extracted and standardized the following features from the PROTAC-DB and PROTAC-Pedia datasets, where a specific combination of the following features corresponds to one experiment: the PROTAC, cell type, E3 ligase, POI, and degradation metrics ($pDC_{50}$ and $D_{max}$).

Each dataset entry includes the SMILES representation of the PROTAC molecule, which we canonicalized and removed stereochemistry information from for consistency. In PROTAC-DB, cell type information was predominantly found in textual assay descriptions, such as "degradation in LNCaP cells after 6 h at 0.1/1000/10000 nM," with "LNCaP" being the cell type in this statement. Cell type information was thus extracted using a named entity recognition (NER) system based on a BERT Transformer (Lee et al., 2020). E3 ligase and POI data was extracted as text. Here, we chose to treat the E3 ligase as a class label due to its being heavily balanced towards VHL and CRBN; however, for the POI, which has 281 possible unique POI labels and can contain mutations (*e.g.*, gene "BTK C481S", which represents the protein Q06187 with a mutation from C to S at residue 481), we decided to extract POI amino acid sequences rather than using the class label. The POI amino acid sequences were obtained via web scraping from Uniprot (EMBL-EBI, 2023), and gene mutations were applied as needed.

Following the approach of DeepPROTACs, degradation activity was classified into entries as *active* if $pDC_{50} \geq 7$ and $D_{max} \geq 80\%$, and *inactive* otherwise. As mentioned above, PROTAC-Pedia entries used a slightly less conservative definition of degradation activity, with $pDC_{50}$ higher than 5 and $D_{max}$ higher than 30%. To maintain consistency, we recalculated DeepPROTACs' active/inactive criteria for PROTAC-Pedia entries with available $DC_{50}$ and $D_{max}$ data.

For PROTAC-DB, part of the degradation activity information, *i.e.*, the $\{DC_{50}, D_{max}\}$ pairs, was obtained by splitting entries containing information for the same PROTAC molecule on multiple assays. For the group of entries in PROTAC-DB which reported $DC_{50}$ but lacked $D_{max}$ information, we binned all entries with known $\{DC_{50}, D_{max}\}$ information and assigned the average $D_{max}$ per bin to the entries with missing $D_{max}$ information. For other entries, the missing $DC_{50}$ information was estimated via interpolating on the dose-response curve the degradation percentage values from

the assay descriptions. We discarded interpolated points that showed either too steep curves or a hook effect.

## 5.2 PROPOSED DATASETS

We present a set of 16 ML-ready datasets, each characterized by unique features and parameter choices derived from the curated data sources. Firstly, we considered a range of train/validation split ratios, including 80/20, 90/10, 95/5, and 99/1. In terms of PROTAC-Pedia entries, we explored two strategies: either integrating its data into the main dataset with subsequent partitioning, or maintaining them as a separate test set, ensuring non-overlapping SMILES representations[3] between validation and test sets. Other selected datasets included entries in the training data with estimated $D_{max}$ and $DC_{50}$ information, as it was believed it may enhance the model's ability to capture more relevant molecular features.

Importantly, we rigorously ensured data integrity and balance. We prevented SMILES (PROTAC) overlap between train, validation, and test sets. The validation set was meticulously designed with a 50/50 active/inactive entry ratio to ensure robust model evaluation. For class balance, oversampling and SMILES randomization were applied to minority class entries in the training datasets.

We believe that this comprehensive dataset generation approach allows for a thorough exploration of different data configurations and preprocessing techniques' impact on model performance and generalization in predicting degradation activity.[4]

Finally, we designed a specific PyTorch Dataset class named `ProtacDataset`, for handling each dataset and transforming non-numerical features as needed. In particular, a `ProtacDataset` can be configured to return different SMILES encodings, such as Morgan (Morgan, 1965), MACCS keys (Durant et al., 2002), and path-based molecular fingerprints (Landrum, 2010), and/or tokenized SMILES strings (Wolf et al., 2020), and/or graph representations from PyTorch Geometric (Fey & Lenssen, 2019). For variable-length fingerprints, $\{1024, 2048, 4096\}$-bit length fingerprints were typically explored. Users can also provide tailored classes for preprocessing the other non-numerical features, *i.e.*, E3 ligase class, cell type class, and POI amino acid sequence. By default, `ProtacDataset` generates ordinal encodings of E3 ligase and cell type features, respectively, and a $m$-to-$n$-grams tokenization for the POI sequence.

## 5.3 PROPOSED MACHINE LEARNING MODELS

We propose a general model architecture for predicting degradation activity of PROTAC complexes. This architecture involves joining (either summing or concatenating) embeddings from different model branches, *i.e.*, encoders, each processing distinct features of the complex. These embeddings are then fed into a final head module, typically a linear layer or an MLP module.

Our open-source implementation, based on PyTorch Lightning, is designed to be fully configurable and customizeable to accommodate diverse encoder modules. In our work, we experimented with the following encoder architectures. The integer classes of the E3 ligase and cell type, respectively, and the POI sequence 2-grams are each processed by a linear layer followed by a batch normalization layer and a ReLU activation function.

To generate PROTAC embeddings, we propose various model architectures tailored to specific molecular representations derived from SMILES. For PROTAC encoding via molecular fingerprints, we compute Morgan and MACCS fingerprints from PROTAC SMILES using the RDKit library toolkit. Two MLP models extract PROTAC embeddings from each fingerprint and are then summed together. For PROTAC encoding via molecular graphs, we use the PyTorch Geometric library to create graph encodings from chemical graphs representing PROTAC molecules. A GNN-based sub-model is then used to processes these graph encodings. Lastly, for PROTAC encoding via SMILES strings, we utilize Transformer-based models like BERT, pre-trained on SMILES data. The selected BERT model is fine-tuned for our task, and it produces a suitable PROTAC embedding representation, which is derived from the CLS output head of BERT.

---

[3]We refer to the whole PROTAC SMILES, not the individual PROTAC components, here.
[4]Please refer to Appendix C for a UMAP visualization and analysis of the proposed datasets.

Table 1: Number of curated data extracted from the PROTAC-DB and PROTAC-Pedia datasets.

| Dataset | Total | Known Degradation Activity | Binned-$D_{max}$ | Curve-Fitted $DC_{50}$ | No Activity Info |
|---|---|---|---|---|---|
| PROTAC-DB | 6,674 | 1,113 (16.7%) | 670 (10.0%) | 362 (5.4%) | 4,529 (67.9%) |
| PROTAC-Pedia | 1,203 | 427 (35.5%) | - | - | 776 (64.5%) |

Table 2: Description of the 16 different datasets curated in this work. Note that while all datasets contain PROTAC-DB data, not all include PROTAC-Pedia data. "Sz" indicates *size*, or the number of compounds in that split, and "A/I" indicates the proportions of *active* and *inactive* data.

| Dataset | Train Sz | Train A/I | Val Sz | Val A/I | Test Sz | Test A/I | PROTAC-Pedia as Test |
|---|---|---|---|---|---|---|---|
| 80/20-split-AC | 3,344 | 50.0% / 50.0% | 535 | 49.9% / 50.1% | - | - | N |
| 80/20-split-AC-T | 2,778 | 50.0% / 50.0% | 413 | 45.5% / 54.5% | 256 | 27.3% / 72.7% | Y |
| 80/20-split | 2,196 | 50.0% / 50.0% | 318 | 50.0% / 50.0% | - | - | N |
| 80/20-split-T | 1,630 | 50.0% / 50.0% | 232 | 50.0% / 50.0% | 274 | 25.5% / 74.5% | Y |
| 90/10-split-AC | 3,612 | 50.0% / 50.0% | 268 | 50.0% / 50.0% | - | - | N |
| 90/10-split-AC-T | 3,002 | 50.0% / 50.0% | 225 | 49.8% / 50.2% | 226 | 22.1% / 77.9% | Y |
| 90/10-split | 2,354 | 50.0% / 50.0% | 159 | 49.7% / 50.3% | - | - | N |
| 90/10-split-T | 1,746 | 50.0% / 50.0% | 116 | 50.0% / 50.0% | 251 | 22.7% / 77.3% | Y |
| 95/5-split-AC | 3,746 | 50.0% / 50.0% | 134 | 50.0% / 50.0% | - | - | N |
| 95/5-split-AC-T | 3,116 | 50.0% / 50.0% | 112 | 50.0% / 50.0% | 209 | 18.2% / 81.8% | Y |
| 95/5-split | 2,434 | 50.0% / 50.0% | 79 | 49.4% / 50.6% | - | - | N |
| 95/5-split-T | 1,804 | 50.0% / 50.0% | 58 | 50.0% / 50.0% | 245 | 21.6% / 78.4% | Y |
| 99/1-split-AC | 3,852 | 50.0% / 50.0% | 27 | 48.1% / 51.9% | - | - | N |
| 99/1-split-AC-T | 3,206 | 50.0% / 50.0% | 22 | 50.0% / 50.0% | 204 | 17.2% / 82.8% | Y |
| 99/1-split | 2,498 | 50.0% / 50.0% | 16 | 50.0% / 50.0% | - | - | N |
| 99/1-split-T | 1,850 | 50.0% / 50.0% | 12 | 50.0% / 50.0% | 240 | 20.8% / 79.2% | Y |

From an empirical analysis, we did not see a significant difference in performance between summing and concatenating embeddings from different branches. As such, we favoured the summing operation as it limits the number of required model parameters and can potentially scale better when eventually introducing additional features encoders. In our experiments, the head of the model, which processes the summed embedding, is a linear layer outputting a single scalar. The model is then trained to minimize a binary cross-entropy loss for predicting whether a given PROTAC complex is active or inactive.

Since the test sets do not present the same percentage of active and inactive entries, *i.e.*, 50-50%, we added a "dummy model" to evaluate the models performance on the test set. Such dummy model always returns the most common class in the given dataset, being it either active or inactive. Ideally, models trained on the train sets shall strive to outperform the dummy model test scores.

### 5.4 HYPERPARAMETER TUNING

Deep learning models involve numerous hyperparameters that significantly impact their performance. Yet, identifying the ideal values is challenging due to the absence of a clear link between combinations of values and performance. To address this, hyperparameter tuning automates the search for optimal values, maximizing (or minimizing, as needed) a predefined performance metric. In our work, we expanded PyTorch Lightning's command line interface (CLI) to integrate hyperparameter tuning using the Optuna framework.

In our experimental setup, we used a TPE sampler together with an Hyperband pruner, while varying the number of trials according to the number of hyperparameters to be tuned per model. We setup a tailored set of tunable hyperparameters for each of the different SMILES encoder surrogate models described in the previous section. More details on the models' hyperparameters and their value range can be viewed in Appendix D.

## 6 RESULTS

### 6.1 DATA CURATION

Table 1 presents an overview of curated data extracted from the PROTAC-DB and PROTAC-Pedia datasets. PROTAC-DB includes a total of 6,674 entries. Among these, 1,113 (16.7%) have known

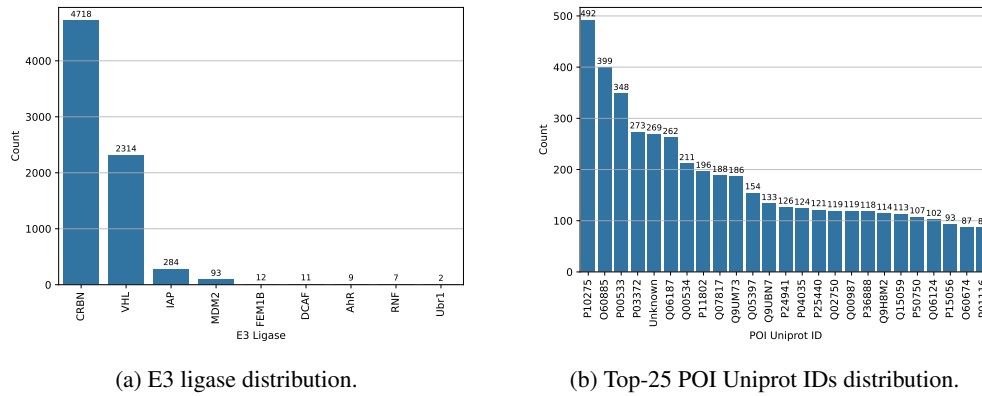

(a) E3 ligase distribution.

(b) Top-25 POI Uniprot IDs distribution.

Figure 2: Distribution of E3 ligase and POI Uniprot IDs for PROTAC-DB and PROTAC-Pedia datasets after data curation.

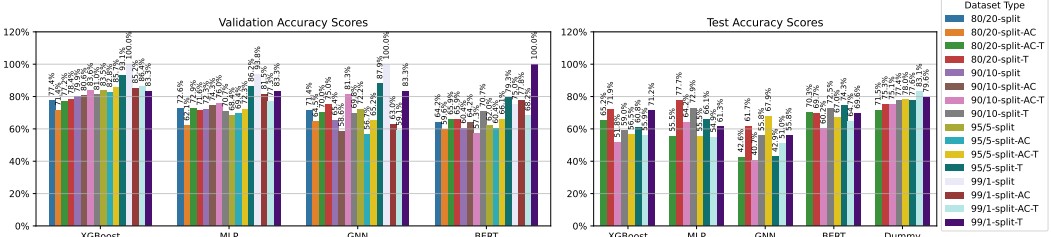

Figure 3: Accuracy scores for degradation activity prediction.

(measured) values for degradation activity, *e.g.*, known $DC_{50}$ and $D_{max}$ values. Another 670 (10.0%) entries have average-binned $D_{max}$ values, and 362 (5.4%) have been curve-fitted to extract $D_{max}$ and $DC_{50}$ values. Notably, a significant portion, 4,529 entries (67.9%), lack activity information. On the other hand, PROTAC-Pedia contains 1,203 entries, with 427 (35.5%) entries having known $DC_{50}$ and $D_{max}$ values. However, a substantial majority of entries in PROTAC-Pedia (776, or 64.5%) do not contain any activity information.

Following the aforementioned methodology for data curation (Section 5.1, Figure 1a), the resulting data were then organized into 16 ML-ready datasets, whose characteristics are summarized in Table 2. Datasets including augmented data in the train set, *i.e.*, average-binned $D_{max}$ and curve-fitted $DC_{50}$, are marked as "AC", whereas datasets including PROTAC-Pedia entries as a separate test set are labeled as "T".

Figure 2a provides insight into the distribution of E3 ligases in the curated dataset, with a total of nine unique E3 ligases identified. The distribution of E3 ligases reveals a significant variation in their frequencies within the datasets, with CRBN and VHL being the most prevalent, while others such as Ubr1 are notably rare.

Figure 2b offers an overview of the distribution of the top 25 most common POI Uniprot IDs within the curated dataset. Among the 281 unique Uniprot IDs present, P10275, the androgen receptor, is the most prevalent, occurring 492 times. Notably, 269 instances are labeled as "Unknown," indicating Uniprot IDs that have not been reported in the original datasets. This distribution highlights the varying frequencies of these specific Uniprot IDs within the dataset, providing insights into the prominence of certain proteins of interest in the analyzed data.

## 6.2 PROTEIN DEGRADATION PREDICTION

The left side of Figure 3 displays validation accuracy results for various model types across the different datasets. For the 80/20-split dataset, the MLP model achieved the highest accuracy at 72.6%, while BERT and GNN models achieved 64.2% and 71.4%, respectively. With additional

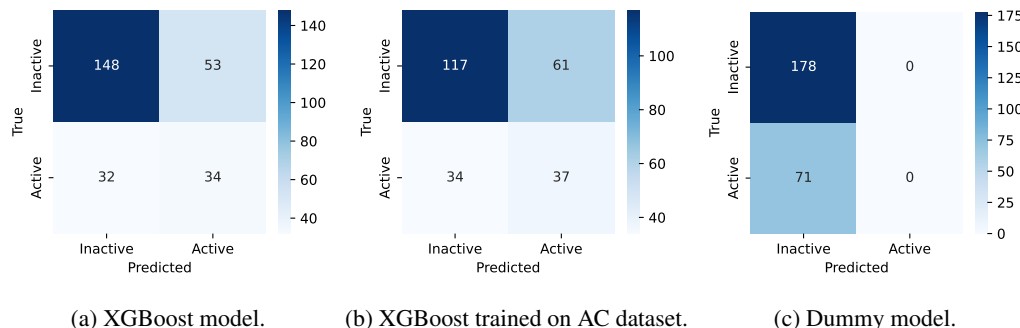

(a) XGBoost model.  (b) XGBoost trained on AC dataset.  (c) Dummy model.

Figure 4: Confusion matrixes for the dummy model and the XGBoost models on the test set of the 80/20 split dataset.

augmentations (AC and AC-T), the performance generally decreases, with the MLP model achieving 62.1% accuracy in the 80/20-split-AC dataset.

When considering test accuracy, in the right side of Figure 3, the performance trends are similar to the validation set. Reasons for the seemingly high performance of the dummy model are discussed in Section 7.3. For the 80/20-split-AC dataset, which includes the largest validation set and is close in size to DeepPROTACs validation set[5], XGBoost achieved the best validation accuracy of 71.4% and a ROC-AUC of 0.73. For the 80/20-split-AC-T dataset, the GNN model achieved the lowest test accuracy at 42.6%, while the BERT model achieved the highest test accuracy at 70.3%.

The performance of a baseline XGBoost model is also included, showing competitive results in both validation and test sets across different dataset splits and augmentations. A full list of performance scores, *i.e.*, F1 score, precision, recall, and ROC-AUC, is reported in Appendix A.

## 7 DISCUSSION

### 7.1 MODEL COMPLEXITY/ACCURACY TRADE-OFF

Here we present various models which achieve comparable performance to DeepPROTACs on certain datasets, and possible outperformance on others. Nevertheless, the models presented herein require a much simpler molecular representation in order to achieve a comparable accuracy on protein degradation predictions, highlighting the strengths of the model in improved protein degradation prediction accuracy with less computational complexity than existing approaches.

### 7.2 THE LIMITATIONS OF 2D REPRESENTATIONS

Unfortunately, it appears that all proposed models reach an upper bound in protein degradation prediction accuracy with the current molecular, protein, and cell representations used, despite extensive hyperparameter optimization and models tried. This suggests that the limitation is not in the model itself, but rather in the representation used, and that moving to more complex molecular representations (*e.g.*, 3D structures) and cell representations (*e.g.*, learned representations) could significantly improve the model performance. Degradation activity will be dependent on cell type, but currently this is only weakly factored in through a one-hot representation.

### 7.3 HIGH TEST ACCURACY OF THE DUMMY MODEL

In order to understand why the test accuracy of the "dummy" model, which simply selects the most popular class label, is the highest among all models in this work, we can look at its confusion matrix. Figure 4 shows the confusion matrix for the test predictions of the dummy model versus the XGBoost models, which scored the highest average F1 score both on the validation and test set (0.82 and 0.36,

---

[5]We did not have access to DeepPROTACs validation data. Li et al. (2022) evaluated their model using a validation set of 567 entries.

respectively). XGBoost models are able to correctly classify only a portion of the active entries, which are the minority class in the test sets (the test sets are unbalanced), thus leading to a lower accuracy compared to the dummy model. On the other hand, the dummy model misclassifies all active entries, always receiving an F1 score of 0 (see Figure 5 in Appendix A).

## 7.4 DATA AUGMENTATION

An unexpected observation in this work was that data augmentation (via the use of mean $D_{max}$ for data points missing $D_{max}$ and curve-fitted $DC_{50}$ for entries missing $DC_{50}$) generally made the models worse in all cases. Another potential avenue to explore is better strategies for data imputation. For instance, could using the median $D_{max}$ value have led to more effective data augmentation?

## 7.5 COMPARISON OF VARIOUS MODEL ARCHITECTURES

The different models investigated had different strengths. For instance, despite the lack of any benefit due to data augmentation, the XGBoost model was at least robust to it (at any data split ratio), while all other models got worse as a result of data augmentation.

Similarly, the BERT-based models showed less accuracy drop between the validation and test sets, suggesting this could be a model which generalizes better relative to the others. Otherwise, the average validation accuracy for the models investigated here-in was as follows: 83.60% for XGBoost, 76.28% for MLP, 73.96% for GNN, and 71.67% for BERT. On the other hand, the average test accuracy was: 62.53% for XGBoost, 63.51% for MLP, 52.29% for GNN, and 68.53% for BERT.

## 7.6 UNDERSTANDING OOD-GENERALIZATION

In order to better understand how the model would fare for in-distribution (ID) and out-of-distribution (OOD) generalization, we investigated the performance across different validation/test splits as well as different split rations. Here, we observed that as the size of the training set increased, the ID validation accuracy also increased. However, the OOD test accuracy did not see the same increases with increasing training set size, in most cases hovering around 50-80% for all models. The test accuracy only increased uniformly with training set size for the dummy model.

Furthermore, when going from the ID (validation) to OOD (test) set F1 score, we saw a mean drop in F1 score of 0.39 (0.73 average validation F1 score and 0.34 average test F1 score), with the drop being most prominent for the XGBoost models (0.46 drop in F1 score) and least for BERT models (0.31 drop in F1 score). All models F1 scores are reported in Appendix A, Figure 5.

Another way to investigate (and potentially improve) OOD generalization is through the design of better dataset splits, which avoid, for instance, the presence of certain POIs in both the train and test sets (generalization to new POIs), or the presence of identical linkers in both the train and test sets (generalization to new linkers). This would guarantee a more accurate estimate of generalization accuracy than random splitting alone and is better reflective of the real use-case scenario of such a model, where it can be used in the scoring function for a PROTAC generative model. Nevertheless, here it was not possible to do it for lack of sufficient and balanced data, such that it could not be done reliably without leaving imbalanced data splits.

## 8 CONCLUSIONS

In this work, we curated open-source PROTAC data and introduced a versatile toolkit for predicting PROTAC degradation effectiveness. Our approach, which leverages machine learning techniques, offers open-source accessibility and ease of reproducibility. The performance of our models, achieving a 71.4% validation accuracy and a 0.73 ROC-AUC validation score compared to the state-of-the-art, demonstrates their competitiveness with existing methods for protein degradation prediction. Furthermore, our toolkit offers a less computationally complex alternative, making it a valuable resource for researchers in molecular drug design. Overall, this work may accelerate advancements in PROTAC engineering

## REPRODUCIBILITY STATEMENT AND CODE AVAILABILITY

Anonymized source code for this work is available at `https://anonymous.4open.science/r/protac-toolkit`. The anonymized repository also contains detailed instructions to reproduce the results presented in this work, including: dataset curation process and the curated datasets, hyperparameter tuning, and the YAML configurations which can be used to train and therefore retrieve all the models we discussed in our research.

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

# A    PREDICTION SCORES

This section provides a collection of validation and test scores of the evaluated models on the proposed datasets. Figures 5, 8, 7, and 6 report validation and test F1, precision, recall, and ROC-AUC scores, respectively, of the evaluated models in this work.

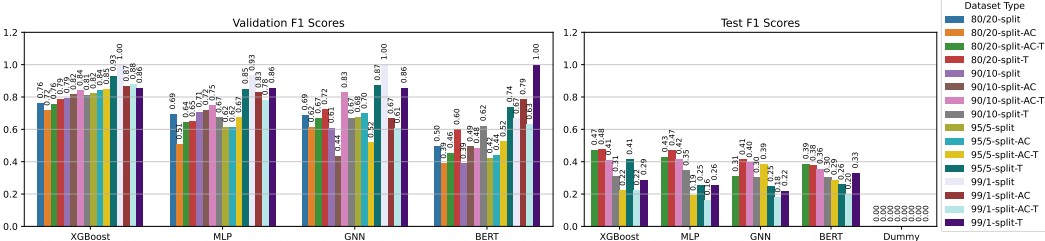

Figure 5: F1 score for predicting activity degradation.

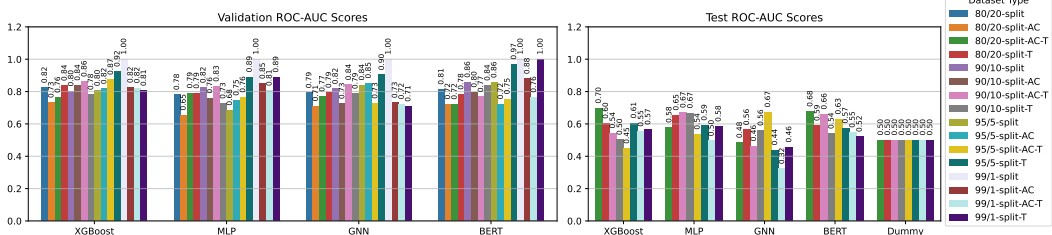

Figure 6: ROC-AUC for predicting activity degradation.

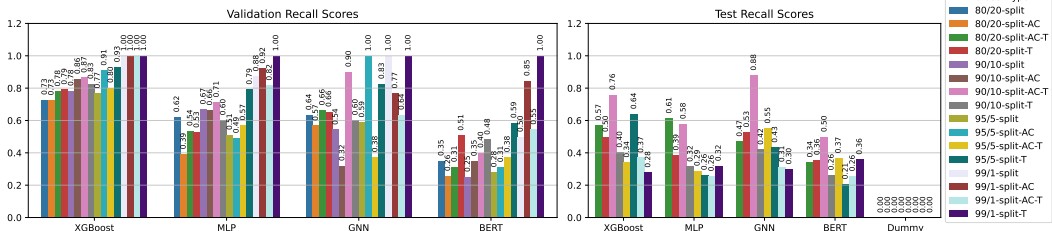

Figure 7: Recall for predicting activity degradation.

# B    XGBOOST AND DUMMY MODELS CONFUSION MATRIXES

This section reports the comparison of the confusion matrixes on the test sets between the XGBoost model versus the dummy model. In particular, Figures 9, 10, and 11 show the confusion matrixes at 90/10, 95/5, and 99/1 train/validation splits, respectively.

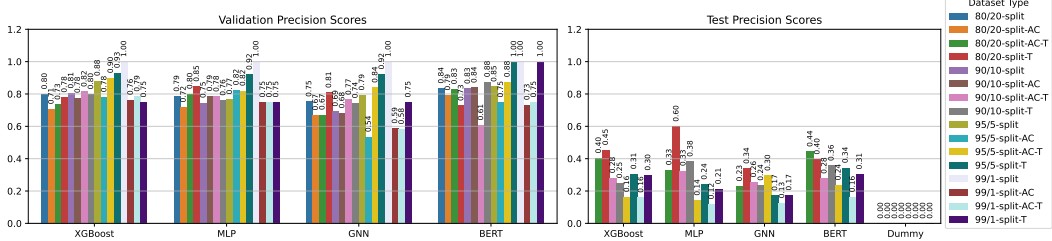

Figure 8: Precision for predicting activity degradation.

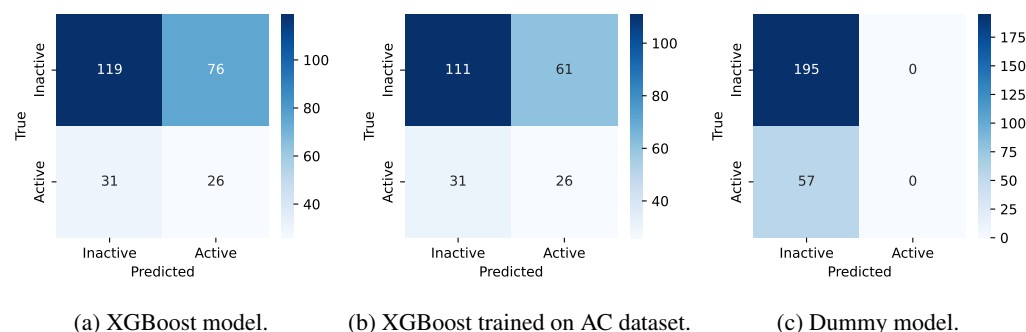

(a) XGBoost model.      (b) XGBoost trained on AC dataset.      (c) Dummy model.

Figure 9: Confusion matrixes for the dummy model and the XGBoost models on the test set of the 90/10 split dataset.

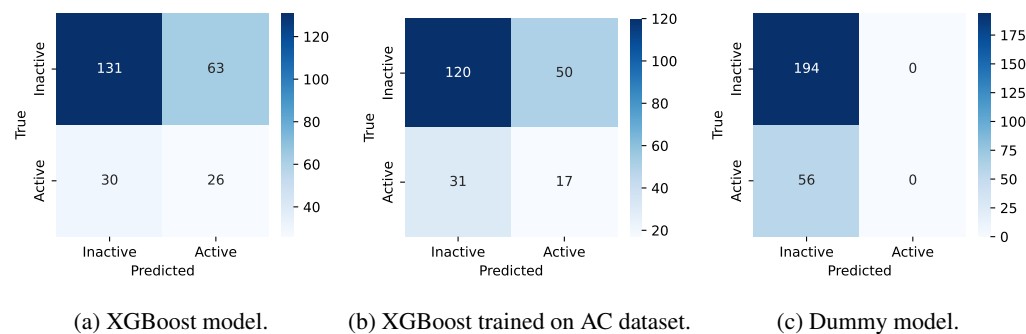

(a) XGBoost model.      (b) XGBoost trained on AC dataset.      (c) Dummy model.

Figure 10: Confusion matrixes for the dummy model and the XGBoost models on the test set of the 95/5 split dataset.

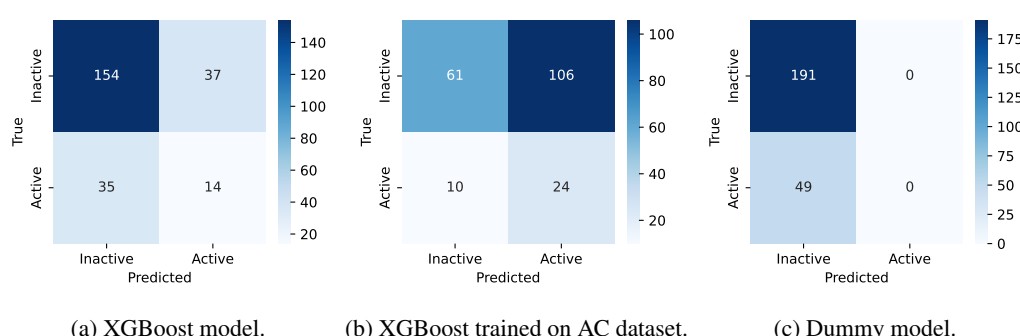

(a) XGBoost model.      (b) XGBoost trained on AC dataset.      (c) Dummy model.

Figure 11: Confusion matrixes for the dummy model and the XGBoost models on the test set of the 99/1 split dataset.

# C UMAP VISUALIZATION AND ANALYSIS OF THE PROPOSED DATASETS

Figure 12 shows the UMAP embeddings of our proposed datasets. The UMAPs where generated from concatenating PROTACs Morgan 2048bit fingerprints with an atomic radius of 6, together with the other numerical features. We can see that samples across train/val/test sets tend to sparsely overlap with each other. However, no clear cluster structures appear overall.

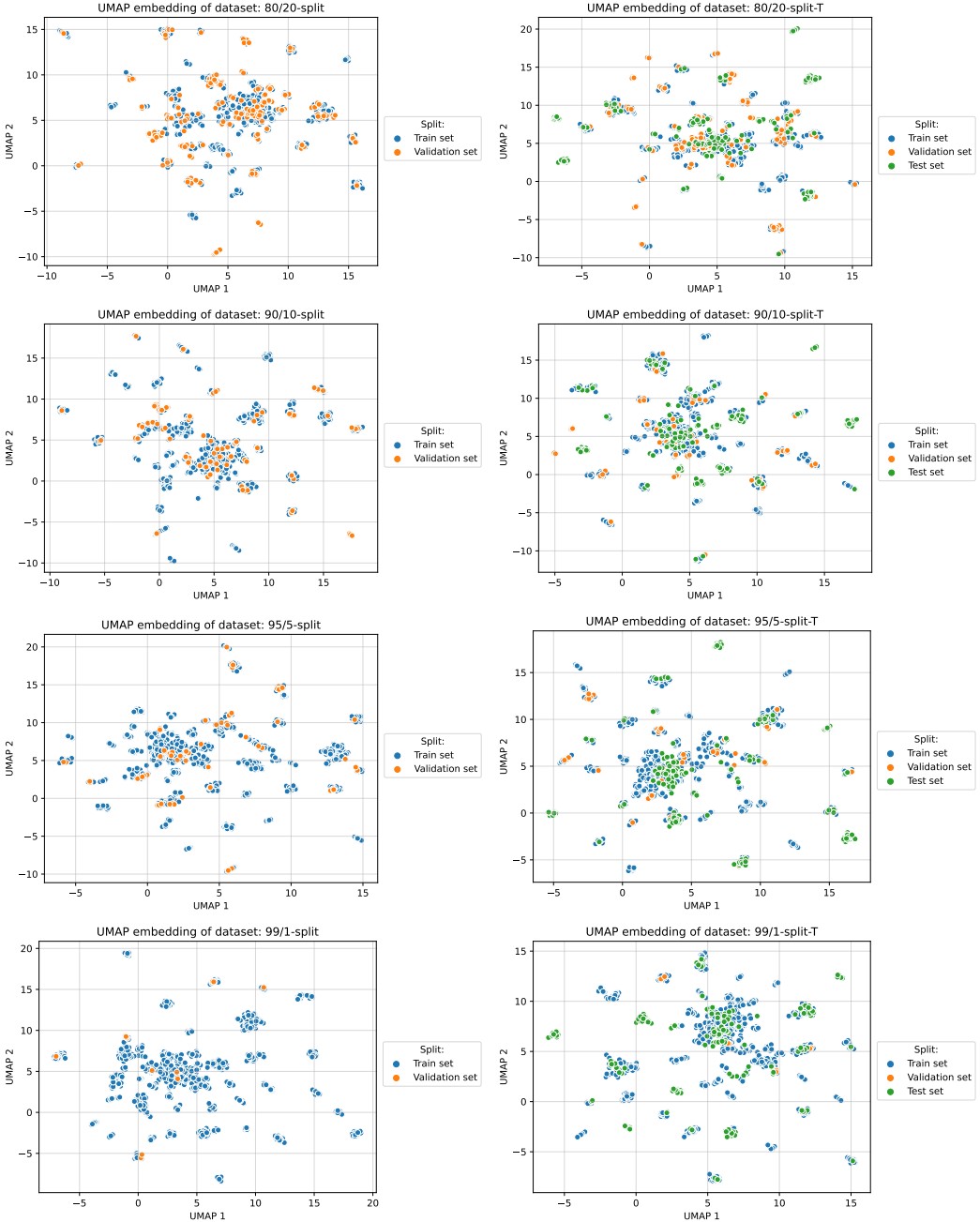

Figure 12: UMAP embeddings of the proposed datasets.

## D HYPERPARAMETERS RANGES FOR OPTUNA OPTIMIZATION

This section lists the configurations of the hyperparameter values to be optimized by Optuna. For XGBoost models, Table 3 include all XGBoost hyperparameters ranges (manually set parameters are marked as "fixed"). For each of the deep learning models types proposed, we report three tables: a table for the fixed hyperparameters, *e.g.*, the number of training epochs and optimizer parameters, another table for Optuna configurations, *e.g.*, the number of trials to run, and a table showing the hyperparameters to be suggested in each trial, together with their range.

Tables 4, 5, and 6 display such configuration information for MLP-based models. Tables 7, 8, and 9 show the aforementioned configuration information for GNN-based models. Finally, Tables 10, 11, and 12 include the configuration information for BERT-based models.

Table 3: XGBoost models hyperparameters.

| Hyperparameter | Range |
|---|---|
| **Fingerprint-specific** | |
| fp_bits | {1024, 2048, 4096} (fixed) |
| fp_type | morgan_fp (fixed) |
| fp_radius | [2, 10] |
| fp_max_path | [8, 10] |
| **XGBoost-specific** | |
| booster | {gbtree, gblinear, dart} |
| lambda | [1e-8, 1.0] (log scale) |
| alpha | [1e-8, 1.0] (log scale) |
| **Additional hyperparameters for booster=gbtree or booster=dart** | |
| max_depth | [1, 16] |
| eta | [1e-8, 1.0] (log scale) |
| gamma | [1e-8, 1.0] (log scale) |
| grow_policy | {depthwise, lossguide} |
| **Additional hyperparameters for booster=dart** | |
| sample_type | {uniform, weighted} |
| normalize_type | {tree, forest} |
| rate_drop | [1e-8, 1.0] (log scale) |
| skip_drop | [1e-8, 1.0] (log scale) |
| **Additional hyperparameters (conditional)** | |
| fp_use_extra_features | {True, False} |

Table 4: Fixed experiment configuration for MLP-based models.

| Parameter | Value |
|---|---|
| Seed | 42 |
| **Trainer Configuration** | |
| Precision | 16-mixed |
| Callbacks | EarlyStopping: val_loss (min), val_acc (max) |
| Max Epochs | 20 |
| Accumulate Grad Batches | 1 |
| Gradient Clip Val | 1.0 |
| **Model Configuration** | |
| Smiles Encoder | RDKitFingerprintEncoder |
| Poi Seq Encoder | POISequenceEncoder |
| E3 Ligase Encoder | E3LigaseEncoder |
| Cell Type Encoder | CellTypeEncoder |
| Head | Linear |
| Join Branches | sum |
| **Data Configuration** | |
| Protac Dataset Args | Use MACCS FP, |
| | Use Morgan FP, |
| **Optimizer Configuration** | |
| Optimizer | AdamW |
| Betas | (0.9, 0.999) |
| Eps | 1.0e-08 |
| Weight Decay | 0.01 |
| Amsgrad | false |

Table 5: Optuna configuration for MLP-based models.

| Hyperparameter | Value / Range |
|---|---|
| n_trials | 100 |
| metric | val_acc |
| study.direction | maximize |
| study.pruner.class_path | optuna.pruners.HyperbandPruner |
| study.pruner.init_args.min_resource | 2 |
| study.pruner.init_args.max_resource | Equal to the number of training epochs |
| study.pruner.init_args.reduction_factor | 3 |
| study.sampler.class_path | optuna.samplers.TPESampler |
| study.sampler.init_args.seed | 42 |

Table 6: Hyperparameter suggestions for Optuna optimization for MLP-based models.

| Hyperparameter | Suggestion Function | Range / Choices |
|---|---|---|
| model.smiles_encoder.init_args.fp_bits | suggest_categorical | [1024, 2048, 4096] |
| data.protac_dataset_args.morgan_atomic_radius | suggest_int | [2, 8] (step=2) |
| model.smiles_encoder.init_args.hidden_channels | suggest_categorical | See choices in the corresponding config file |
| data.batch_size | suggest_int | [64, 256] (step=64) |
| optimizer.init_args.lr | suggest_float | [1e-5, 1e-3] (log scale) |

Table 7: Fixed experiment configuration for GNN-based models.

| Parameter | Value |
|---|---|
| Seed | 42 |
| **Trainer Configuration** | |
| Precision | 16-mixed |
| Callbacks | EarlyStopping: val_loss (min), val_acc (max) |
| Max Epochs | 20 |
| Accumulate Grad Batches | 1 |
| Gradient Clip Val | 1.0 |
| **Model Configuration** | |
| Smiles Encoder | GnnSubModel |
| Poi Seq Encoder | POISequenceEncoder |
| E3 Ligase Encoder | E3LigaseEncoder |
| Cell Type Encoder | CellTypeEncoder |
| Head | Linear |
| Join Branches | sum |
| **Data Configuration** | |
| Protac Dataset Args | Include Smiles as Graphs, |
| | Precompute Smiles as Graphs, |
| **Optimizer Configuration** | |
| Optimizer | AdamW |
| Betas | (0.9, 0.999) |
| Eps | 1.0e-08 |
| Weight Decay | 0.01 |
| Amsgrad | false |

Table 8: Optuna configuration for GNN-based models.

| Hyperparameter | Value / Range |
|---|---|
| n_trials | 50 |
| metric | val_acc |
| study.direction | maximize |
| study.pruner.class_path | optuna.pruners.HyperbandPruner |
| study.pruner.init_args.min_resource | 2 |
| study.pruner.init_args.max_resource | Equal to the number of training epochs |
| study.pruner.init_args.reduction_factor | 3 |
| study.sampler.class_path | optuna.samplers.TPESampler |
| study.sampler.init_args.seed | 42 |

Table 9: Hyperparameter suggestions for Optuna optimization for GNN.

| Hyperparameter | Suggestion Function | Range / Choices |
|---|---|---|
| model.smiles_encoder.init_args.model_type | suggest_categorical | [gin, gat, gcn, attentivefp] |
| model.smiles_encoder.init_args.jk | suggest_categorical | [last, cat] |
| model.smiles_encoder.init_args.out_channels | suggest_int | [64, 256] (step=64) |
| model.smiles_encoder.init_args.num_layers | suggest_int | [16, 32] (step=8) |
| data.batch_size | suggest_int | [64, 256] (step=64) |
| optimizer.init_args.lr | suggest_float | [1e-5, 1e-3] (log scale) |

Table 10: Fixed experiment configuration for BERT-based models.

| Parameter | Value |
|---|---|
| Seed | 42 |
| **Trainer Configuration** | |
| Precision | 16-mixed |
| Callbacks | EarlyStopping: val_loss (min), val_acc (max) |
| Max Epochs | 20 |
| Accumulate Grad Batches | 4 |
| Gradient Clip Val | 1.0 |
| **Model Configuration** | |
| Smiles Encoder | TransformerSubModel |
| Poi Seq Encoder | POISequenceEncoder |
| E3 Ligase Encoder | E3LigaseEncoder |
| Cell Type Encoder | CellTypeEncoder |
| Head | Linear |
| Join Branches | sum |
| **Data Configuration** | |
| Protac Dataset Args | Smiles Tokenizer: seyonec/ChemBERTa-zinc-base-v1 |
| **Optimizer Configuration** | |
| Optimizer | AdamW |
| Learning Rate | 5e-5 |
| Betas | (0.9, 0.999) |
| Eps | 1.0e-08 |
| Weight Decay | 0.01 |
| Amsgrad | false |

Table 11: Optuna configuration for BERT-based models.

| Hyperparameter | Value / Range |
|---|---|
| n_trials | 5 |
| metric | val_acc |
| study.direction | maximize |
| study.pruner.class_path | optuna.pruners.HyperbandPruner |
| study.pruner.init_args.min_resource | 2 |
| study.pruner.init_args.max_resource | Equal to the number of training epochs |
| study.pruner.init_args.reduction_factor | 3 |
| study.sampler.class_path | optuna.samplers.TPESampler |
| study.sampler.init_args.seed | 42 |

Table 12: Hyperparameter suggestions for Optuna optimization for BERT-based models.

| Hyperparameter | Suggestion Function | Range / Choices |
|---|---|---|
| data.batch_size | suggest_int | [32, 128] (step=32) |

