# OpenReview forum: "Machine Learning for PROTAC Engineering"
_ICLR.cc/2024/Conference — Submitted to ICLR 2024_

### Official Review · Reviewer_qQeP · 2023-11-01

**Soundness:** 2 fair
**Presentation:** 2 fair
**Contribution:** 2 fair
**Rating:** 5
**Confidence:** 3

**Summary:**

In this paper, the author proposed and developed an architecture powered with deep learning to predict protein degradation. The suggested architecture consists of assembling embeddings from various model branches each processing distinct features.

**Strengths:**

- The steps around data collection and data curation of the PROTAC data were really well defined and explained.
- Really detailed litterature review by mentioning the various work that were applied on PROTAC data.
- Validation on different types of data splits.

**Weaknesses:**

The authors did not clearly mention and explain many experimental details that are important to reproduce the reached results.

**Questions:**

- At the end of the "Related Work" (section 4) the author mentioned: "Thirdly, the model’s performance is not benchmarked against other SOTA models with high predictive capacity, such as LightGBM or XGBoost" by state of the art here does the authors refer specificly to PROTAC research or for general ML and DL applications? since claiming that LightGBM and XGBoost are the best as predictive models is too generic and is not true with the presence of different advanced neural network architectures like transformers and attention models.

 - In section 5.2, the authors mentioned about the oversampling that they applied on the minoriy class and I quote: "For class balance, oversampling and SMILES randomization were applied to minority class entries in the training datasets." however they did not specify the type of oversampling/data augmentation algorithm or logic that was leveraged for this purpose, was it some random duplication? SMOTE oversampling algorithm? and if any what is the reason behind using that specific oversampling method.

 - At the beginning of section 5.3, the author mentioned the following:" We propose a general model architecture for predicting degradation activity of PROTAC complexes. This architecture involves joining (either summing or concatenating) embeddings from different model branches". So did the authors applied a suming or a concatenation to assemble the different embeddings?

 - Still in section 5.3, the authors talked about how they generated the embeddings derived from SMILES, they mentioned the approach with MLP, GNN and BERT, However later on in the experimental comparison they compared those three to XGBoost, so was XGBoost also used to extract representations from SMILES, if so how was it done? If not what is the reason behind comparing it to MLP, GNN and BERT?

 - I do not quite understand the reason behind writing section 7.3, isn't it somehow obvious that a dummy model will have a good accuracy but a bad F1-score since it naively classifies the majority class?

- Concerning section 7.6 about Out-of-Distribution (OOD) generalization, could the authors provide the amount of domain shift happening between the train and test data? This would be helpful to quantitatively grasp the gap between the 2 sets and thus observe how generalizable is the approach towards OOD.

---

### Official Review · Reviewer_BNfR · 2023-11-02

**Soundness:** 3 good
**Presentation:** 2 fair
**Contribution:** 2 fair
**Rating:** 5
**Confidence:** 4

**Summary:**

This paper introduces a benchmark for assessing PROTAC molecules' degradation efficacy. The authors' key contributions include:

-   The collation and organization of PROTAC data from existing databases into a set of 16 comprehensive datasets featuring parameters such as pDC50, Dmax, E3 ligase type, and POI sequence.
-   The creation of an open-source toolkit using PyTorch designed for PROTAC data analysis, featuring a custom dataset class, various models, and a system for optimizing hyperparameters.
-   The evaluation of several foundational models including MLP, GNN, and BERT to predict PROTAC activity, with the best-performing model achieving a validation accuracy of 71.4%.

**Strengths:**

**Originality**: The provision of a new open toolkit for PROTAC modeling is noted. However, the utilized model architectures and molecular representations are established techniques in drug discovery. A comparative discussion on how this study's data curation advances beyond existing methodologies such as those in DeepPROTACs would be beneficial.

**Quality**: The dataset curation and model implementation demonstrate rigor, but the evaluation lacks a thorough examination of generalizability to out-of-distribution data. The validation accuracy is on par with certain benchmarks but does not reach the latest advancements in PROTAC prediction.

**Clarity**: The manuscript is articulate in method and result description. Nonetheless, it could benefit from a more detailed discussion on dataset balancing and optimization strategies. The section on limitations requires expansion to provide a deeper insight.

**Significance**: The toolkit has the potential to facilitate open PROTAC research; however, its impact might be limited by the conventional nature of the modeling techniques employed. The real value for the field appears to be in the datasets provided, despite the datasets being variations of the same data compilation.

The paper, as it stands, may not meet the innovation and impact criteria for a premier ML conference. Enhancements in methodology and evaluation could be considered to aim for such venues. Currently, it is more suited to a workshop setting.

**Weaknesses:**

**Rigor**: The benchmarking methodology should include comparisons with leading-edge models like DeepPROTACs. Clarity on whether these benchmarks are evaluated on identical test splits would be informative.

**Technical Depth**: The choice of 2D molecular representations might be constraining accuracy. The consideration of 3D structural representations may provide accuracy benefits. Additionally, the simplistic approach to embedding combination could overlook essential interaction details. Incorporating explicit modeling of PROTAC-target engagement may be advantageous.

**Organization**: The first four sections display redundancy, with overlap in the introduction, background, and contributions. The scientific context of PROTAC, while informative, could be condensed or moved to an appendix to make room for more technical details of the current contribution.

**Questions:**

**A/I Balance**: The discrepancy in the active/inactive data proportions across training, validation, and test splits, specifically the ~20%/80% in the test split, raises questions regarding the choice of this distribution. How would the baseline and control (i.e., "dummy") models perform under a balanced test split? A uniform distribution across all splits reflecting the overall dataset composition would likely yield a more fair and robust evaluation framework.

---

### Official Review · Reviewer_A2WQ · 2023-11-05

**Soundness:** 1 poor
**Presentation:** 2 fair
**Contribution:** 1 poor
**Rating:** 3
**Confidence:** 4

**Summary:**

The paper presents a strategy for curating open-source PROTAC data and proposes an open-source toolkit for predicting the degradation effectiveness of novel PROTAC molecules. PROTACs are a novel class of therapeutic agents that can degrade specific proteins by recruiting an E3 ubiquitin ligase and a target protein of interest (POI). The authors collect and standardize data from two existing open-source datasets, PROTAC-DB and PROTAC-Pedia, which contain structural and experimental data of PROTAC complexes. The paper organizes the curated data into 16 datasets incorporating essential features such as pDC50, Dmax, E3 ligase type, POI amino acid sequence, and experimental cell type. This work provides an open-source toolkit that includes a configurable PyTorch dataset class, a customizable machine-learning model, and a hyperparameter optimization mechanism. The toolkit allows users to process various PROTAC features and train different models for protein degradation prediction. Three surrogate models that utilize different PROTAC representations are developed, such as molecular fingerprints, molecular graphs, and SMILES strings. The paper evaluates the models on the public datasets and compares them with existing state-of-the-art models. The paper claims that the proposed models achieve comparable or better performance with less computational complexity and more reproducibility.

**Strengths:**

The paper presents a strategy for curating open-source PROTAC data and proposes an open-source toolkit for predicting the degradation effectiveness of novel PROTAC molecules. The paper also develops three surrogate models that utilize different PROTAC representations, such as molecular fingerprints, molecular graphs, and SMILES strings. The proposed models achieve comparable or better performance with less computational complexity and more reproducibility compared to existing state-of-the-art models. The paper’s approach combines existing ideas and applications to a new domain.

The authors collect and standardize data from two open-source datasets, PROTAC-DB and PROTAC-Pedia, which contain structural and experimental data of PROTAC complexes. The paper organizes the curated data into 16 different datasets incorporating important features such as pDC50, Dmax, E3 ligase type, POI amino acid sequence, and experimental cell type. The paper provides an open-source toolkit that includes a configurable PyTorch dataset class, a customizable machine-learning model, and a hyperparameter optimization mechanism. The toolkit allows users to process various PROTAC features and train different models for protein degradation prediction.

The paper is well-written and easy to follow. The authors clearly explain the concepts and methods used in the study. The paper includes detailed descriptions of the datasets, models, and experiments conducted. The authors also provide visualizations of the results to aid in understanding.

**Weaknesses:**

The paper evaluates the proposed models on automatically curated public datasets and compares them with existing state-of-the-art models. However, the curation performed does not account for the dimension of time, which is critical metadata for protac datasets as pDC50 and Dmax can vary substantially based on the timepoint due to variations in parameters such as the kinetics of degradation and protein resynthesis rate, etc. The authors should explicitly account for time in their modeling framework.

In the curve fitting section of the data curation, the authors use a standard four-parameter fit using the Hill Equation, similar to what is commonly used with equilibrium pharmacology. This type of fit does not accurately estimate DC50 or Dmax because of the hook effect. A fit similar to that proposed by Haid et al (https://doi.org/10.3390/pharmaceutics15010195) is preferred.

The paper could benefit from additional experiments that evaluate the models on more diverse and challenging datasets, such as those specific to a congeneric series or those with different E3 ligase types and POI amino acid sequences.

The work focuses on predicting protein degradation efficacy using machine learning models. However, the paper does not address other important aspects of PROTAC engineering, such as pharmacokinetics, toxicity, or other ADMET properties. The paper could benefit from additional experiments that address these aspects and provide a more comprehensive view of PROTAC engineering.

**Questions:**

Why do the authors remove stereochemistry information in their molecular standardization workflow? In many instances one stereoisomer may be significantly more potent than another, which becomes especially apparent in degrader design. The authors should show how their modeling framework is affected by retaining stereochemical information vs. removing it.

---

### Author Response · Authors · 2023-11-22
**Response to Reviewer A2WQ**

## Response to Reviewer A2WQ

Thank you for your insightful comments and valuable feedback. We appreciate your recognition of the strengths of our work, particularly in developing an open-source toolkit for PROTAC data analysis and the creation of three surrogate models. We would like to address the concerns and questions you raised:

1. **Time Dimension in PROTAC Datasets:**
   While we acknowledge the importance of time as a factor in PROTAC datasets, our current work focuses on the available data, which currently does not include time metadata. We also align our approach with existing methodologies in the field, such as the DeepPROTACs work, which does not incorporate time information either. We believe our work still provides significant insights and lays a foundation for future studies that could incorporate time data when it becomes readily available.

2. **Curve Fitting and the Hook Effect:**
   We appreciate the suggestion regarding the curve fitting approach. We chose the Hill Equation for its common use and applicability in equilibrium pharmacology. However, we understand the concern about the hook effect and its impact on accurately estimating DC50 or Dmax. We addressed this by discarding entries that exhibited the hook effect, ensuring the integrity and relevance of our data. We describe this in the text in section 5.1 *Data Curation*, at the top of page 5. Future work could explore alternative fitting methods, such as the one proposed by Haid et al., to enhance the accuracy of these estimates.

3. **Additional Experiments and Datasets:**
   We agree that evaluating our models on more diverse datasets could further demonstrate their robustness. Our focus on openly available datasets was a deliberate choice to promote reproducibility and accessibility. Future iterations of our work could indeed benefit from exploring more challenging and closed-source datasets, including those containing different E3 ligase types and POI sequences.

4. **Other Aspects of PROTAC Engineering:**
   Our current focus was on the prediction of protein degradation efficacy using machine learning models. While aspects such as pharmacokinetics and toxicity are undoubtedly important in PROTAC engineering, they were beyond the scope of this study. However, we believe our toolkit has the potential to be included in broader generalization pipelines, addressing various aspects of PROTAC engineering in future work.

5. **Stereochemistry Information:**
   The decision to remove stereochemistry information was driven by practical considerations. In laboratory settings, testing all chiral versions of a compound is a common approach, regardless of which stereoisomer is predicted to be active. We aimed to reflect this practicality in our modeling framework by not explicitly modeling stereochemistry. In *in silico* ADMET modeling, it is not important to identify the most active stereoisomer, but rather what the highest activity out of all possible stereoisomers would be. On top of that, only one of our proposed models (BERT-based) is currently able to leverage such additional information about the data. As such, removing stereochemistry information created a common and general foundation for all our models. However, we acknowledge the importance of stereochemistry in drug design and agree that future studies could benefit from retaining and analyzing this information on the performance of more advanced models.

We hope these responses adequately address the concerns raised and illustrate our commitment to advancing the field of PROTAC engineering with robust, reproducible, and accessible tools and methodologies.

---

### Author Response · Authors · 2023-11-22
**Response to Reviewer BNfR**

## Response to Reviewer BNfR

Thank you for the thoughtful evaluation of our paper. We appreciate the recognition of the originality and quality of our work, particularly in the dataset curation and toolkit development. We would like to address the following concerns and questions:

1. **Benchmarking Methodology:**
   We acknowledge the importance of comparing our models with previously-published models like DeepPROTACs. However, it is important to note that the DeepPROTACs model is not reproducible from the available information in the publication, nor are their training and evaluation datasets openly available, limiting our ability to make a direct comparison. Our focus on reproducibility and simplicity positions our work as a necessary step for future comparisons with more complex models, including DeepPROTACs.

2. **Technical Depth and Model Choices:**
   Our choice of 2D molecular representations was driven by the balance between model complexity and performance. We observed that our models reached an upper bound in accuracy with the current representations, suggesting that more complex models (e.g., using 3D data) might not necessarily yield better results. This finding aligns with our goal to demonstrate the efficacy of simpler models before progressing to more complex ones.

3. **Manuscript Organization:**
   We appreciate the feedback on the organization of the manuscript. We will consider restructuring the content to reduce redundancy and provide more technical details about our current contribution.

4. **A/I Balance and Model Evaluation:**
   The active/inactive data proportions were chosen to reflect the real-world distribution of PROTAC datasets. However, we understand the concern regarding the test split distribution. In future work, we may explore how the models perform under a balanced test split to provide a more comprehensive evaluation framework.

We hope these responses address the concerns raised, and that they demonstrate our commitment to advancing the field of PROTAC research through accessible and robust methodologies.

---

### Author Response · Authors · 2023-11-22
**Response to Reviewer qQeP**

## Response to Reviewer qQeP

Thank you for your detailed review of our paper. We appreciate the recognition of the strengths in our data collection and curation processes. We would like to address your questions and concerns:

1. **State-of-the-Art Models Reference:**
   Our reference to LightGBM and XGBoost as state-of-the-art models pertains to their application in general machine learning tasks, rather than being specific to PROTAC research. In many low-data domains, tree-based models can perform better than neural architectures despite being cheaper to train, and we thus felt it important to make this comparison. We acknowledge the advancements in neural network architectures like transformers and attention models and believe that our work provides a necessary comparison point to these more complex models.

2. **Oversampling Technique:**
    We used random sampling of the minority class. Original SMOTE was not a viable option because two of our models require SMILES data in order to obtain their inputs, and we wanted a general solution for comparing all the models. On top of that, from a UMAP analysis, we noticed that there is not a clear decision boundary and that most entries across dataset splits are anyway overlapping. Therefore, we speculated that other variations of SMOTE that sample data over the decision boundaries would most likely collapse (or generate very similar solutions) to a random sampling strategy.

3. **Embedding Assembly in Model Architecture:**
   In our architecture, we show results of embeddings being summed together. From our previous work and empirical analysis, we did not see a significant difference in performance between summing and concatenating embeddings from different branches. As such, we favoured the summing operation as it limits the number of required model parameters and can potentially scale better when eventually introducing additional features encoders.

4. **Comparison with XGBoost:**
   The comparison with XGBoost was intended to demonstrate the efficacy of simpler machine learning models against more complex deep learning architectures. In our work, XGBoost served as a benchmark to evaluate the necessity and efficiency of employing deep learning models for this task.

5. **Section on Dummy Model and OOD Generalization:**
   The inclusion of the dummy model's performance was to provide a baseline for comparison and to highlight the challenges in achieving high F1-scores with imbalanced datasets, as also mentioned at the end of Section 5.3 _Proposed Machine Learning Models_. Regarding OOD generalization, we acknowledge the importance of quantifying domain shifts between train and test data. Future work can include a more detailed analysis of this aspect to better understand the generalizability of our approach.

We hope these responses clarify our methodologies and address the raised concerns, and we are committed to continuously refining our work to contribute meaningfully to the field of PROTAC research.

---

### Meta-Review · Area_Chair_gFU5 · 2023-12-10

**Metareview:**

The paper focuses on predicting PROTAC degradation efficacy using machine learning models. The reviewers have pointed out several areas of improvement. One of the main criticisms is that the models were evaluated on automatically curated public datasets without accounting for the dimension of time, which is a critical metadata for PROTAC datasets. The authors are also critiqued for using a standard four-parameter fit using the Hill Equation in the curve fitting section of the data curation, which does not accurately estimate DC50 or Dmax. The reviewers suggest using a fit similar to that proposed by Haid et al. Additionally, the reviewers feel that the paper could benefit from additional experiments that evaluate the models on more diverse and challenging datasets. They also recommend that the authors address other important aspects of PROTAC engineering such as pharmacokinetics, toxicity, and other ADMET properties.

**Justification For Why Not Higher Score:**

All reviewers have expressed concerns about the paper. Although the authors have responded to these issues, many of the queries have been deferred to future research, and no modifications or enhancements have been made to the current submission.

**Justification For Why Not Lower Score:**

N/A

---

### Decision · Program_Chairs · 2024-01-16

Reject